# Effect of Hybrid Filler, Carbon Black–Lignocellulose, on Fire Hazard Reduction, including PAHs and PCDDs/Fs of Natural Rubber Composites

**DOI:** 10.3390/polym15081975

**Published:** 2023-04-21

**Authors:** Przemysław Rybiński, Ulugbek Zakirovich Mirkhodjaev, Witold Żukowski, Dariusz Bradło, Adam Gawlik, Jakub Zamachowski, Monika Żelezik, Marcin Masłowski, Justyna Miedzianowska

**Affiliations:** 1Institute of Chemistry, The Jan Kochanowski University, 25-406 Kielce, Poland; 2Department of Biophysics, National University of Uzbekistan, Tashkent 100095, Uzbekistan; u.z.mirkhodjaev@gmail.com; 3Department of General and Inorganic Chemistry, Cracow University of Technology, 31-155 Cracow, Poland; 4Institute of Geography and Environmental Sciences, Jan Kochanowski University, 25-406 Kielce, Poland; 5Institute of Polymer and Dye Technology, Faculty of Chemistry, Lodz University of Technology, 90-924 Lodz, Poland

**Keywords:** NR rubber, natural filler, fire hazard, smoke density, polychlorinated dibenzo-p-dioxins and furans, polycyclic aromatic hydrocarbons, toxicometric index

## Abstract

The smoke emitted during thermal decomposition of elastomeric composites contains a significant number of carcinogenic and mutagenic compounds from the group of polycyclic aromatic hydrocarbons, PAHs, as well as polychlorinated dibenzo-p-dioxins and furans, PCDDs/Fs. By replacing carbon black with a specific amount of lignocellulose filler, we noticeably reduced the fire hazard caused by elastomeric composites. The lignocellulose filler reduced the parameters associated with the flammability of the tested composites, decreased the smoke emission, and limited the toxicity of gaseous decomposition products expressed as a toximetric indicator and the sum of PAHs and PCDDs/Fs. The natural filler also reduced emission of gases that constitute the basis for determination of the value of the toximetric indicator W_LC50SM_. The flammability and optical density of the smoke were determined in accordance with the applicable European standards, with the use of a cone calorimeter and a chamber for smoke optical density tests. PCDD/F and PAH were determined using the GCMS-MS technique. The toximetric indicator was determined using the FB-FTIR method (fluidised bed reactor and the infrared spectrum analysis).

## 1. Introduction

In recent years, we have observed a significant increase in popularity of polymer composites containing fillers of natural origins. The increased use of biocomponents in the production of composite materials is a result of, on the one hand, legal regulations related to materials fit for re-use and recycling, and, on the other hand, the necessity to reduce the costs of their production. Polymer composites containing fillers of natural origins, obtained on the basis of thermoplastic materials or chemically hardened resins, are currently commonly used in many industries, e.g., automotive, packaging, and means of transport [1,2,3,4,5].

Unlike thermoplastic materials or resins, use of natural fillers in the case of elastomers, which are a specific group of macromolecular plastics, is very limited.

Elastomers are a group of polymers characterised by a high elastic deformation event at very low stresses. The main representative of this group of plastics is rubber. It is made of caoutchouc plants, mainly *Hevea brasiliensis* [6,7,8,9].

In order to obtain satisfactory mechanical parameters, elastomeric composites must contain a reinforcing filler, most commonly in the form of carbon black. The carbon black content in elastomeric mixtures is often as high as 100 phr. However, while carbon black improves the mechanical properties of composites, it negatively impacts their fire hazard parameters, especially the amount of smoke during their thermal decomposition [10,11,12,13].

Generally, we may assume that the value of parameter SD_MAX_ (maximum smoke optical density) increases proportionally with the content of carbon black in the elastomeric composite.

It should be clearly emphasised that smoke not only makes it harder for people to navigate in rooms on fire, but also, being a part of smog caused by, e.g., uncontrolled burning of rubber waste in household fireplaces, it creates a lethal threat for people as it absorbs carcinogenic and mutagenic compounds [14].

We know that the smoke emitted during decomposition of elastomeric composites contains a significant number of carcinogenic and mutagenic compounds from the group of polycyclic aromatic hydrocarbons, PAHs, as well as polychlorinated dibenzo-p-dioxins and furans, PCDDs/Fs.

The PAH group compounds, especially benzo(a)pyrene, are known to be carcinogenic [15,16].

The toxic effect of the PCDD/F group compounds mainly consists of slow but steady damaging of dividing cells in living organisms. These substances, due to the fact that they damage the DNA code, are mutagens. Their teratogenic and allergic effects have been proven. There are documented cases of serious skin allergies caused by dioxins, so called chloracne.

During scientific research conducted on laboratory animals, mainly rats and mice, a high increase in liver and lung neoplastic tissues was determined [17,18]. It should be emphasised that the toxic effect of dioxins on living organisms should always be considered from the point of view of their synergy with other contaminations, such as PAHs, the carcinogenic effect of which is beyond doubt, or nitric oxides, sulphur oxides, carbon oxides, as well as hydrogen chloride, hydrogen bromide, or hydrogen cyanide (toximetric indicator) [19,20]. Therefore, the toxic effect of dioxins should be understood as complex processes of damaging of internal organs, painful allergic rashes, as well as mutagenic, teratogenic, or carcinogenic effects. The most disturbing toxic effect of dioxins is probably damaging of the structure of the genetic code contained in DNA.

This paper shows that by replacing carbon black with a specific amount of lignocellulose filler in the form of cereal straw, as well as Miscanthus or beech wood, it is possible to noticeably reduce the fire hazard caused by elastomeric composites. The lignocellulose filler reduced the parameters associated with their flammability, decreased the smoke emissions, and limited the toxicity of gaseous decomposition products expressed as a toximetric indicator and the sum of PAHs and PCDDs/Fs.

The article also shows that substitution of a specific, defined amount of carbon black with a lignocellulose filler does not limit the mechanical parameters expressed as tear resistance, TS, and elongation at break, as well as Eb of the tested elastomeric composites.

## 2. Methods

### 2.1. Materials

The lignocellulose filler in the form of beech wood, straw, and Miscanthus were obtained from a sawmill and local farms, respectively. Miscanthus is a perennial grass that is a mixture of Chinese Miscanthus (Miscanthus sinensis) and sugar (Miscanthus sacchariflorus). After drying in a laboratory dryer (70 °C, 24 h), filler in the form of beech wood, straw, or Miscanthus was ground into a fine powder using a ball mill Pulverisette 5 Classic Line planetary ball mill (Fritsch, Idar-Oberstein, Germany) for 3 h.

Natural rubber (Torimex-Chemiclas Ltd., Lodz, Poland) (NR) in the form of ribbed smoke sheets (RSS1) was used as the elastomer matrix. The elastomeric composites were cross-linked with the use of sulphur (Siarkopol, Poland) (S, in the quantity of 2 phr per 100 phr of rubber) in the presence of mercaptobenzothiazole (Saint Louis, MO, USA) (MBT, in the amount of 2 phr per 100 phr of rubber), stearic acid (Avantor Performance Materials, Gliwice Poland) (SA, in the quantity of 1 phr per 100 phr of rubber), and zinc oxide (Huta Będzin, Poland) (ZnO, in the quantity of 5 phr per 100 phr of rubber). The compositions of the elastomer mixtures are presented in Table 1.

Carbon black, HAF 330 (CB) (Makrochem, Poland) was used as a black filler.

Melamine polyphosphate (MPP) (Everkem, Italy) was used as a flame retardant.

### 2.2. Preparation of Composites

The process of making rubber blends was completed in two stages. In the first stage, NR was plasticized (4 min) and mixed with the filler (4 min). The mixing was carried out using a laboratory blender (Brabender, Duisburg, Germany) at 50 °C with a rotational speed of 40 rpm. In the second stage, the system of vulcanizing components was mixed in using laboratory rolling mills (Table 1).

The rubber blends were vulcanized in steel moulds placed between electrically heated press shelves. The optimal vulcanization time (*τ*_0.9_) at a temperature of 160 °C was determined by means of a WG-2 vulcameter according to PN-ISO 3417:1994.

### 2.3. Determination of Natural Filler Distribution in Polymer Matrix

The dispersions of the lignocellulose filler in the NR rubber matrix were investigated using atomic force microscopy (AFM). AFM measurements were performed using a Metrology Series 2000 apparatus Molecular Imaging (North Miami, FL, USA). Imaging was performed with the use of a scanning siliceous head with conical shape (dilation angle < 20°) and height of about 15–20 μm operating in oscillatory mode with a resonance frequency of about 170 kHz. The samples used for measurements were vulcanized in a steel mould. A glass plate was placed in the mould to obtain a low coarseness for the vulcanizate surface. Before placing in the mould, the glass plate surface was rinsed with acetone and dried with an air jet to degrease it and remove impurities.

The image analysis was performed using the the WS × M program developed and made available by Horcas et al. [21].

### 2.4. Cone Calorimeter

Flammability tests were conducted with the use of a cone calorimeter from Fire Testing Technology Limited. Nanocomposites samples with dimensions of 100 × 100 ± 1 mm and a thickness of 20 ± 0.5 mm were tested in a horizontal position, with a density of heat radiation flux amounting to 35 kW × m^−2^ [22].

### 2.5. Optical Smoke Density

The optical smoke density was determined with the use of a smoke density chamber produced by the Fire Testing Technology. Samples with dimensions of 75 mm × 75 mm and thickness of 2 mm were tested in a horizontal position with a heat flux density of 25 kW/m^2^ without the use of a gas burner (non-flammable tests). The tests were conducted in accordance with the ISO 5659 standard [10].

## 3. Determination of PCDDs/Fs and PAHs

### 3.1. Sampling

Elastomeric composites were burned in a smoke-forming test chamber. For the analysis of each PCDDs/Fs and benzo(a)pyrene (representative compounds for PAHs), the total amount of combustion products in 1 m^3^ of air was used. ORBO1000 (Supelco) samplers were used for sampling. When sampling for the determination of PCDDs/Fs, it contained PUR foam measuring 22 mm × 87 mm. For sampling for the determination of polycyclic aromatic hydrocarbons, the samplers contained a composite filling: PUR foam 22 mm × 30 mm, XAD-2 resin in the amount of 1.5 g, and PUR foam 20 mm × 30 mm [10].

### 3.2. Determination of PCDDs/Fs

Polychlorinated dibenzodioxins and polychlorinated dibenzofurans were extracted from PUR foam using ultrasound-assisted extraction. The extract was evaporated to dryness, and the analytes were dissolved in 2 mL of n-hexane. The extract was purified on a multi-layer Silicon Oxide column (Multi-layer Sillica Gel Dioxin Column Supelco) and a Dual-layer Carbon Reversible Tube Supelco. The analytes were eluted with 50 mL of toluene and dissolved in 100 μL of dodecane after evaporation to dryness.

Polychlorinated dibenzodioxins were determined by isotopic dilution using a gas chromatograph coupled to a triple quadrupole mass spectrometer dedicated to the determination of dioxins (GCMS 8050 NX with Schimadzu DIOXIN PACKAGE). The separation of polychlorinated dibenzodioxin congeners and polychlorinated dibenzofurans was carried out on an SI-5Sil MS 60 m × 0.25 mm column with a film size of 0.25 μm. The calculation was performed using LABSolution Insight GC. The uncertainty of the μ measurements was expressed as the extended uncertainty for k = 2 at a confidence level of 95%. Limit of quantification: LOQ—0.1 pg/g of the sample [10].

### 3.3. Determination of PAH

PAHs were extracted from XAD-2 resin and PUR foam with 100 mL acetone with using the ultrasound-assisted extraction technique. The extracts were evaporated to dryness and in the next step the analytes were dissolved in 1 mL of organic solvent (n-hexane). The extracts were purified using the SPE technique on Chromabond SiOH 6 mL, 500 mg columns. The analyte was eluted with a mixture of n-hexane and dichloromethane in a ratio of 25:75. After evaporation to dryness, the analyte was dissolved in 100 μL of dichloromethane and filtered using 13 mm syringe filters with a PTFE membrane. PAHs were determined using an external standard and a gas chromatograph coupled to a triple quadrupole mass spectrometer (GCMS 8050 NX Schimadzu).

The separation was carried out on the HT8-PCB MS column 60 m × 0.25 mm with a film size of 0.25 um (SGE GC column). The calculation was performed in LABSolution Insight GC. The uncertainty of the μ measurements was expressed as the extended uncertainty for k = 2 at a confidence level of 95%. Limit of quantification: LOQ—0.1 ng/g of the sample [10].

### 3.4. Determination of Toxicometric Indicators

A fluidised bed reactor was used to determine the quantity and quality of emitted gaseous products during thermal decomposition of the studied composites (Figure 1). The procedure of analysis was conducted according to the PN-88-B-02855 standard. The procedure of the investigation was described previously [23,24,25].

### 3.5. Mechanical Properties

The measurement of the mechanical properties was performed in accordance with the PN-ISO 37:1998 standard using a Zwick machine, model 1435.

The tests included: stress at elongation 100%, 200%, 300% (SE100, SE200, SE300), tensile strength (TS), and relative elongation at break (Eb). Each determination was performed for five samples. The test was carried out at a constant speed of 500 mm/min.

## 4. Results

### 4.1. Morphology of Studied Composites

The AFM analysis indicated that the cellulose filler both in the form of beech wood, straw, or Miscanthus disperses in the NR rubber matrix very well (Figure 2). All fillers in the form of CW as well as CM and CS did not show an excessive tendency for agglomeration in the polymer matrix, which results from both weak polymer filler and filler–filler interactions, as well as from relatively large particles in the filler itself. The average diameter of CW, CM, and CS is 6 (CW) and 725 (CM and CS) [10].

The results obtained with the use of the AFM method indicate that the CW was characterized by the most homogenous distribution in the polymer matrix. Both CM and CS also underwent homogenous distribution in the polymer matrix, but in the case of CM, and especially CS, little agglomerates were observed (Figure 2).

### 4.2. Fire Hazard

#### 4.2.1. Flammability

Carbon black is the most popular filler used in the elastomeric industry. It is composed of 90–94% C, 0.1–8% O, and 0.2–1% H and has the crystallographic structure of graphite. The surface of carbon black, similar to surfaces of degraded polycyclic aromatic hydrocarbons with different degrees of oxidation, is characterized by high reactivity. The surface of soot may include both groups that are electron acceptors, e.g., carbonyl, carboxyl, hydroxyl, anhydride, or lactone groups, as well as groups that are electron donors, such as the γ structure of pyrone. Generally, the surface of energetically heterogeneous soot is dominated by carbonyl groups; nevertheless, the pH value of carbon black used in the rubber industry is greater than 7, which results from the absorption of various chemical substances with alkaline natures on its surface [26,27,28].

The impact of carbon black on the flammability of composites that contain it is ambiguous. On the one hand, one should consider that carbon black is a source of pure carbon, and thus it should catalyse the process of combustion of a composite that contains it; on the other hand, by immobilising macromolecules of elastomer on its surface, it decreases the amplitude of their thermal vibrations, and as a result, limits the process of thermal decomposition of the composite. The higher stability of elastomers filled with carbon black also results from the fact that carbon black sweeps the macroradicals created during the thermal degradation of the elastomer composite. On the surface of the majority of types of carbon black, one can observe a high concentration of paramagnetic centres, which are donors of chemically stable radicals.

A composite that includes carbon black in the amount of 20 phr is characterised by the values of parameters HRR_MAX_ (maximum heat release rate), THR (total heat released), AMLR (average mass loss rate), and MARHE (maximum average rate of heat emission) at the level of 402.5 kW/m^2^, 30.8 MJ/m^2^, 19.26 g/m^2^×s, and 163.8 kW/m^2^, respectively (Table 2).

Substitution of a part of carbon black with a lignocellulose filler in a form of straw (CS), Miscanthus (CM), or beech wood (CW) reduced the fire hazard parameters of the investigated composites. It should be clearly emphasised that an increase in the content of the lignocellulose component in the elastomeric mixture, especially in the form of beech wood, significantly reduced the fire hazard parameters. Composite C-6, which contains CW:CB filler in a weight ratio of 2:1, was characterised by a reduction of parameters HRR_MAX_, THR, AMLR, and MAHRE by 17.1%, 12%, 13.3%, and 15.07%, respectively, in relation to reference composite C-1. When the contents of the lignocellulose filler in the form of lignocellulose wood increased, the values of parameters HRR_MAX_, THR, AMLR, and MAHRE decreased. For example, composite C-7 with a CW:CB filler weight ratio of 5:1, had values of parameters HRR_MAX_, THR, AMLR, and MAHRE that were 37.9%, 12.3%, 31.5%, and 38.7% lower, respectively, in relation to the reference composite C-1.

Moreover, a lignocellulose filler in the form of both cereal straw (CS) and Miscanthus (CM) significantly reduced the value of the fire hazard parameters (Table 2, Figure 3).

According to the literature, the thermal decomposition of lignocellulose raw materials highly depends on their chemical composition. Even a small amount of lignin (10%) increases their resistance to fire [29]. Tests conducted by Kozłowski and associates clearly indicated that natural raw materials rich in lignin (linen and hemp) are characterised by a lower rate of heat release compared to raw materials with a high cellulose content. This confirms lignin’s ability to catalyse carbonisation reactions that positively impact the creation of an insulating carbon layer, which increases the fire resistance of elastomer biocomposites [30,31,32,33].

The flammability tests (Table 2) clearly indicated that the composites containing beech wood were characterised by significantly reduced flammability compared to composites filled with cereal straw or Miscanthus. The differences in the fire hazard parameters were directly related to the contents of lignin in the natural raw materials. The content of lignin in beech wood ranges from 20 to 35% of its weight; in the case of ripe cereal straw, the amount is 15–20%, while in Miscanthus, depending on its age (Miscanthus harvested after 1 year, 2 years, or 3 years), the content of lignin varies in the range of 10–20% of its weight.

The flammability of elastomeric composites containing a natural filler is also impacted by cellulose. Currently, it is believed that raw materials with a high content of cellulose are characterized by a high level of crystallinity and, consequently, a high amount of flammable levoglucosan is generated during their decomposition, which increases its susceptibility to fire. On the other hand, a significant level of crystallinity of cellulose indicates a high value for its activation energy of destruction, which can, for instance, increase the value of its flash point (ignition temperature). Moreover, the crystalline structure of cellulose is related to a high level of ordering (orientation) of fibrils. The higher the degree of orientation is, the lower amount of oxygen that can diffuse into the fibre [32].

According to Chapple and Anandjiwala, from the point of view of the flammability of composite materials, natural fillers should be characterised by a lower degree of crystallinity, a high level of polymerisation, as well as the highest possible degree of orientation of fibrils [31].

Composites that contain a natural filler were also filled with 10 phr of melamine polyphosphate. Melamine polyphosphate works both in the gaseous phase and the solid phase, decreasing the flammability of the composite that contains it. During the heating of a composite, before its ignition, MPP easily sublimates by absorbing and then releasing heat outside the sample. During thermal decomposition, especially at the flame burning stage, melamine polyphosphate releases significant amounts of chemically inert gases, i.e., nitrogen, ammonia, and nitric oxides, which, by reducing the concentration of oxygen in the environment of the reaction, reduce the flame temperature. However, the essential activity of melamine polyphosphate consists of the creation of an intumescent boundary layer between the flame and the sample, which efficiently limits the transport of mass and energy. According to the literature, in order create a carbonised boundary layer, apart from an MPP compound that acts as an acidic foaming factor, a carbon donor, usually pentaerythritol, is necessary [34]. As a result of esterification (condensation) reactions between MPP and PER hydroxyl groups, a foamed, carbonised boundary layer is created, which efficiently limits the transfer of mass and energy between the sample and flame.

The conducted research clearly indicates that the introduction of PER into the NR rubber composite matrix containing MPP did not noticeably reduce their flammability.

In light of the above, the foamed boundary layer generated during decomposition of composites C-2 ÷ C-7 was considered to result from the reaction of the condensation of MPP hydroxyl groups with the hydroxyl groups of the cellulose filler, which also plays the role of a carbon donor in the composite.

#### 4.2.2. Smoke Emission

Smoke emission has a significant impact on the fire hazard caused by elastomeric composites.

Smoke is defined as an aerosol of solid or liquid particles, generated during the process of pyrolysis or thermal and oxidation decomposition of organic fuel. The amount of generated smoke depends on many factors, which include: the fuel’s chemical composition, availability of oxygen, intensity of the impact of heat stream, as well as the conditions of the combustion process (flame-less or flame combustion) [35,36].

The amount of smoke generated during the thermal decomposition of a polymer depends mainly on the chemical structure of macromolecule. For example, thermal decomposition of polymethylmethacrylate (PMMA), as over 90% of it depolymerises to methyl methacrylate, i.e., a compound of high oxygen content (esters group), is nearly emission-free, while PVC emits large amounts of black, toxic smoke during combustion [37,38].

This happens due to the generation of a significant amount of hydrogen chloride and the creation of unsaturated carbon structures at the initial stage of thermal decomposition of PVC, which, as a result of subsequent reactions, are subject to cyclisation and aromatisation. We currently know that aromatic destructs, because they condense in the gaseous phase in flames to polyaromatic compounds, are perfect precursors of soot, and as a consequence, they increase the amount of emitted smoke.

The amount of soot, the main component of smoke, generated by organic compounds, changes in the following sequence: naphthalenes, benzene rings, diolefins, monoolefins, and parafine.

The mechanism of the generation of smoke is very complex. It is assumed that in the first stage, as a result of the thermal decomposition of polymeric materials, low molecular weight destructs are produced, which, depending on the chemical structure of the material, may be in the form of nitriles, amines, cyclic ketones, esters, alcohols, as well as unsaturated compounds, including cyclic compounds. The main precursors of smoke are unsaturated and aromatic carbon structures, which, in the gaseous phase at the flame temperature, create reactive, aromatic transitional forms, which constitute a nucleus of soot and precursors of bounded aromatic organic compounds from the PAH and PCDD/F groups. Coagulation and agglomeration of soot nuclei result in the production of molecules of soot, which contain absorbed, bounded ring organic compounds on their surfaces (Figure 4).

The results of the smoke emission tests indicate that a lignocellulose filler in the form of cereal straw, Miscanthus, or beech wood limited the amount of smoke generated during the thermal decomposition of the tested composites (Table 3).

It should be noted that, while composite C-2, containing 13.5 phr of cereal straw, was characterised by a greater value of the maximum optical smoke density parameter SD_MAX_ compared to reference composite C-1, the value of its VOF4 parameter was significantly lower in relation to sample C-1. The reduction of the VOF4 value indicates a restriction in the rate of production of smoke due to lignocellulose, especially at the initial stage of thermal decomposition of the composite.

It should also be emphasised that the efficiency of a natural filler in the process of the generation of smoke is directly proportional to its content in the composite. The greatest reduction in the values of parameters SD_MAX_ and VOF4 was obtained, regardless of the type of lignocellulose, for 16.5 phr of natural filler in the composite. The lower smoke emission of composites containing lignocellulose in relation to the reference composite resulted mainly from a reduction of the polymer component in the composite, as well as from the fact that the natural filler was subjected to combustion rather than high-temperature cyclisation processes (Table 3).

The reduction in the amount of emitted smoke directly translates into a decreased fire risk. It is commonly known that smoke not only make it harder to navigate in rooms on fire, and thus is the main factor that causes disorientation and panic, but it is also a carrier of toxic organic destructs from the dioxin and polycyclic aromatic hydrocarbons groups.

#### 4.2.3. Dioxin/Furan (PCDD/F) and PAH Toxicity

Dioxins include a group of 75 chlorine-based dibenzodioxins (PCDDs) and 135 chlorine-based dibenzofurans (PCDFs). Among this enormous group of compounds, only several of them show very strong toxic properties for humans and animals. Connections with halogen atoms in the PCDD or PCDF molecule at positions 2, 3, 7, and 8 (there are 17 in total) make those compounds very toxic. In particular, the congener 2, 3, 7, 8-tetrachlorodibenzodioxin (2,3,7,8-TCDD) is the most toxic compound in this group. The toxicity of TCDD exceeds the toxicity of potassium cyanide or strychnine by several orders of magnitude. This compound is easily soluble in fats and is characterised by high a stability under ambient conditions. Laboratory tests show that contact with chlorinated and brominated dioxins, even at a concentration one thousand times lower compared to the concentrations of other contaminations in the environment, is very dangerous for people and animals. Dioxins and their derivatives are carcinogenic, cause biological and biochemical changes, and disturb the process of replication of DNA. They negatively impact cells and tissues, affecting their growth rate. TCDD causes significant skin lesions, especially on the face [39,40].

In order to determine the potential toxicity of the tested samples in relation to the content of dioxins in routine chemical analyses, eighteen most toxic PCDDs/PCDFs congeners were designated, as is the standard. Among 210 congeners of PCDDs and PCDFs, only those that contained atoms of chlorine in the positions indicated as 2, 3, 7, and 8 were designated.

Table 4 presents the concentrations of 17 congeners of PCDDs/PCDFs measured in the gaseous destructs produced during decomposition of the reference composite (sample C-1) and composites containing cereal straw (sample C-3), Miscanthus (sample C-5), and beech wood (sample C-7), in a weight ratio with soot of 5:1.

The test results clearly indicated that a cellulose filler significantly reduced the amounts of dioxins produced during thermal decomposition of the composite. The greatest reduction in the total concentration of dioxins, expressed as ng/g of sample, was recorded for composites containing both cereal straw and Miscanthus.

Determination of the level of toxicity of the sample, expressed as TEQ (*Toxic Equivalency*), is performed using the so called TEF (Toxic Equivalency Factor). *TEQ* is calculated using Equation (1) based on the results of the chemical analyses of the mass content of all seventeen PCDD/PCDF congeners. The numerical value of *TEQ* constitutes the total value of partial parameters, obtained after multiplying the analytical result of a concentration of a single PCDD or PCDF congener by the partial factor *TEF*.
(1)$$TEQ=\overset{i=17}{\underset{i=1}{\sum }}\left(m_{i}\times TEF_{i}\right)$$
where:
*m_i_*—the mass of a single congener;*TEF_i_*—the toxicity equivalency factor for the i PCDD/F congener in relation to congener 2, 3, 7, 8-TCDD.


The numerical values of *TEF* are summarised in Table 5. They specify the relative toxicity of each PCDD/PCDF congener in relation to the most toxic congener 2,3,7,8-TCDD, for which, in line with recommendations of the WHO from 1998, factor 1 was adopted. Accordingly, for the least toxic PCDDs/PCDFs, OCDD and OCDF, the TEF factor of 0.0001 was adopted.

*TEQ* is the value that specifies the level of toxicity of the analysed sample in relation to the sum of masses of the mentioned congeners.

The TEF values calculated for the tested composites (Table 6) clearly indicate that composite C-7, which contains a lignocellulose filler in a form of beech wood, was characterised by the lowest toxicity in relation to PCDDs/PCDFs. This suggests that, in the case of composite C-7, the lower emission of compounds from the dioxin and furan groups results from not only the lower content of the polymer component in the composite, but probably also stems from a lower composite combustion temperature value, as well as a reduced amount of generated smoke compared to reference composite C-1.

The introduction of a lignocellulose filler into the matrix of natural rubber also limits the amount of polycyclic aromatic hydrocarbons produced during its thermal decomposition (Table 7).

Polycyclic aromatic hydrocarbons occur in the environment in the form of a mixture of compounds containing two or more aromatic rings in a molecule, ordered in different manners; nevertheless, each two connected rings have two common carbon atoms. Polycyclic aromatic hydrocarbons include many compounds that differ from each other not only in terms of the number of aromatic rings, but also in relation to the positions in which rings are connected with each other, as well as the number and the position of substituents and their chemical properties.

The majority of polycyclic aromatic hydrocarbons show toxic properties. After penetrating an organism, they move to all tissues containing fat, and then they are stored in the fat tissue of the kidneys, liver, and lungs, and are gradually released from there. Polycyclic aromatic hydrocarbons are subject to metabolic conversion in the liver in two phases. The first phase includes reactions leading to the production of compounds that are more active compared to the initial substance. The reactions occurring during this phase include oxidation catalysed by P-450 cytochrome enzymes and hydroxylation with participation of an epoxy hydrolase. Epoxydiols produced in this way are hydrophilic. They participate in the reactions of the second phase, i.e., reactions of binding with endogenic conjunction factors, such as sulphuric acid, glucuronic acid, or glutation.

Active metabolites produced as a result of the initial phase may bind with cell macromolecules (DNA, proteins, and lipids) or generate reactive forms of oxygen. The ability to covalently bind with DNA of a cell is especially inherent to epoxydiols, in which the epoxy group is adjacent to the area of increased electron density of a given hydrocarbon, which translates into the higher chemical and biological activities of this group. The creation of DNA–polycyclic aromatic hydrocarbon adducts may induce pro-mutagenic damage, which is preserved as a form of mutation if it is not removed in by repair processes. Changes in the genetic material may lead to initiation and progression of a neoplastic stage [41].

It is believed that metabolic conversions of polycyclic aromatic hydrocarbons also include the creation of reactive forms of oxygen, e.g., peroxide anion and peroxide, which leads to creation of highly toxic hydroxyl radicals. The products of oxidation of polycyclic aromatic hydrocarbons cause the peroxidation of lipids and disturb the stability of membranes and cellular organelles. During the oxidation of lipids, toxic substances are produced that may have cytotoxic effects. They change the composition of enzymes and disturb their function, cause abnormalities in the processes of transcription and replication, and can increase the number of mutations.

Polycyclic aromatic hydrocarbons may also have an immunotoxic effect. Exposure to these compounds leads to changes in the organs of the immune system: bone marrow, thymus, spleen, and lymph nodes. Moreover, it leads to a reduction in the total number of lymphocytes, eosinophilic granulocytes, immunoglobulins of the IgM and IgA classes, and reduced activity of NK (natural killer) cells.

Polycyclic aromatic hydrocarbons include over 100 substances from five chemical groups: derivatives of anthracene, phenanthrene, chrysene, pyrene, and cholanthrene. Among these, 18 compounds were determined to be toxic (Table 7). Among all polycyclic aromatic hydrocarbons, benzo(a)pyrene is the most researched one, with proven carcinogenic and mutagenic effects [41].

The data included in Table 7 clearly show that a lignocellulose filler significantly reduced not only the total amount of polycyclic aromatic hydrocarbons, but also limited the amount of produced benzo(a)pyrene. The concentration of benzopyrene in gaseous structures of composites C-3, C-5, and C-7 were lower by 83.7%; 89.6%, and 87.5%, respectively, compared to the gaseous destructs of reference composite C-1.

#### 4.2.4. Toxicometric Indicators

In the gaseous products of thermal decomposition of the tested vulcanisates, the amounts of emitted carbon monoxide and dioxide, as well as nitrogen, sulphur, hydrocyanic acid, and hydrogen chloride dioxide, were determined. Based on the obtained emissions, a toxicometric indicator was calculated for each material.

Based on the test results, composite C-1 was characterised by the greatest emission of CO_2_, i.e., approx. 2.8 g/g at 750 °C, as well as the lowest emission of CO_2_ at 450 °C (approx. 1 g/g) among all the samples.

The actual emission of CO demonstrated a slight variability depending on the type of sample as well as the temperature of its decomposition. The highest amount of CO emission, approx. 0.33 g/g, was obtained for composite C-3 at 550 °C. It should be emphasised that all the tested composites had the highest emission of carbon monoxide at the exact temperature of 550 °C, while the highest values were recorded at 750 °C, which is undoubtedly associated with afterburning of CO to CO_2_ (Figure 5).

Sulphur oxide (IV) is a gas which, despite its low actual emissions (below 0.01 g/g), significantly impacts the total toxicity expressed by the toxicometric indicator. This is associated with its low concentration limit, which for SO_2_ is 0.7 g/m^3^ (Table 8). All tested vulcanisates were characterised by the highest emission of SO_2_ at 550 °C, i.e., in the temperature range of the burning of the remnants after thermal decomposition of the sample. It should be clearly emphasised that the composites that contain a lignocellulose filler were characterised by noticeably higher values of emission of SO_2_ compared to the reference composite (C-1). This probably results from the presence of sulphur, mainly in the form of a thiol group in the fillers of natural origins. The largest emission of SO_2_ was recorded in the case of the composite containing beech wood, which may indicate that the percentage value of sulphur is highest in wood compared to cereal straw or Miscanthus (Figure 5).

Hydrocyanic acid, similarly to sulphur oxide, is characterised by a low value of concentration limit, and similar to SO_2_, it significantly impacts the total toxicity of the gaseous components. The test results clearly indicate that the emissions of HCN was significant especially at 550 °C for composites C-3, C-5, and C-7, i.e., for composites containing melamine polyphosphate (Figure 5).

In the case of all tested samples, the emission of hydrogen chloride during their thermal decomposition was below the detection limit.

In accordance with PN-88-B-02855:1988, based on the values of emissions of CO, CO_2_, HCN, HCl, NO_2_, and SO_2_ at 450, 550, and 750 °C, the value of the toxicometric indicator W_LC50SM_ was determined. The value of this indicator constitutes the basis for classification of materials in terms of toxicity of their gaseous destructs. Materials characterised by an indicator W_LC50SM_ < 15 are classified as very toxic materials; those characterised by an indicator W_LC50SM_ within the range of 15–40 are classified as toxic materials; and those >40 are classified as moderately toxic materials [42].

The toxicometric indicator W_LC50SM_ is the arithmetic mean of the indicator W_LC50M_ based on the formula:(2)$$\frac{1}{W_{\mathrm{LC}50M}}=\overset{n}{\underset{i=1}{\sum }}\frac{1}{W_{\mathrm{LC}50,i}}$$

W_LC50_ determines the mass of a given material whose decomposition or combustion under test conditions produces the toxic concentration limit of a given decomposition or combustion product (Table 8) [42]:(3)$$W_{\mathrm{LC}50,i}=\frac{{\mathrm{LC}}_{50,i}}{E_{i}}$$

The data presented in Table 9 show that both composite C-3 and composite C-5, which include a lignocellulose filler in a form of, respectively, cereal straw and Miscanthus, were characterised by the highest values of indicator W_LC50SM_ and thus the lowest toxicity compared to reference composite C-1. The lower toxicity of composites C-3 and C-5 compared to composite C-1 mainly resulted from the lower emissions of CO and CO_2_, especially at 750 °C (the higher value of indicator W_LC50_). Vulcanisate C-7 was characterised by a comparable value of the toxicometric indicator in relation to reference composite C-1.

### 4.3. Mechanical Properties

It is assumed that the main factor that impacts the mechanical parameters of polymer composites is the amount and type of the filler, which shows a tendency to create own structure in the form of three-dimensional spacious network that is able to interact with polymer chains via functional groups. The impact of the type and amount of the filler on the mechanical properties of composites was determined on the basis of the parameters of tear strength (TS), elongation at break (Eb), and stress at elongation 100% (SE100), 200% (SE200), and 300% (SE300) [43].

Based on the test results, it was found that the introduction of a lignocellulose filler to the matrix of NR rubber did not significantly impact the values of parameters TS, EB, SE100, SE200, and SE300. In the case of composites C-3 ÷ C-6, a greater value of the tear strength parameter compared to reference composite C-1 was recorded. The value of the elongation at break parameter for composites C-2, C-4, C-5, and C-7 was higher compared to reference composite C-1 (Table 10).

## 5. Summary

By replacing carbon black with a specific amount of lignocellulose filler in the form of cereal straw, as well as Miscanthus or beech wood, it is possible to noticeably reduce the fire hazard caused by elastomeric composites. It was shown that the flame-retardant effect of raw materials of natural origins was directly associated with their lignin content, which, by catalysing carbonisation, positively impacts the creation of an insulating carbon layer.

The main precursors of smoke are unsaturated and aromatic carbon structures, which in the gaseous phase at the flame temperature, create reactive, aromatic transitional forms, which constitute a nucleus of soot and precursors of bound aromatic organic compounds from the PAH and PCDD/F groups. Coagulation and agglomeration of soot nuclei results in the production of molecules of soot, which contain absorbed, bound ring organic compounds on their surfaces. The results of smoke emission tests indicated that a lignocellulose filler in a form of cereal straw, Miscanthus, or beech wood reduced the amount of smoke generated during the thermal decomposition of the tested composites. The lower smoke emissions of composites containing lignocellulose in relation to the reference composite resulted mainly from the reduction of the polymer component and soot in the composite, as well as from the fact that the natural filler was subjected to combustion rather than high-temperature cyclisation processes.

A lower smoke emission is directly associated with reduction of carcinogenic emissions of organic compounds from the polycyclic aromatic hydrocarbon and PCDD/F groups.

The composites containing natural fillers characterized by 90.5% (CS), 88.7% (CM) and 16.85% (CW) lower values of TEQ in comparison to the reference sample. The natural filler, in the form of CS, CM, and CW, also limited PAH emissions by about 83.38%, 87.36%, and 87.10%, respectively.

The reduced toxicity, expressed by the toxicometric indicator, of vulcanisates containing a natural filler compared to the reference vulcanisate resulted from lower emissions of both CO and CO_2_ at 750 °C.

A natural filler did not negatively impact the mechanical properties of the composites, including their tear strength (TS), elongation at break (Eb), and stress at elongation (SE100-300).

## Figures and Tables

**Figure 1 polymers-15-01975-f001:**
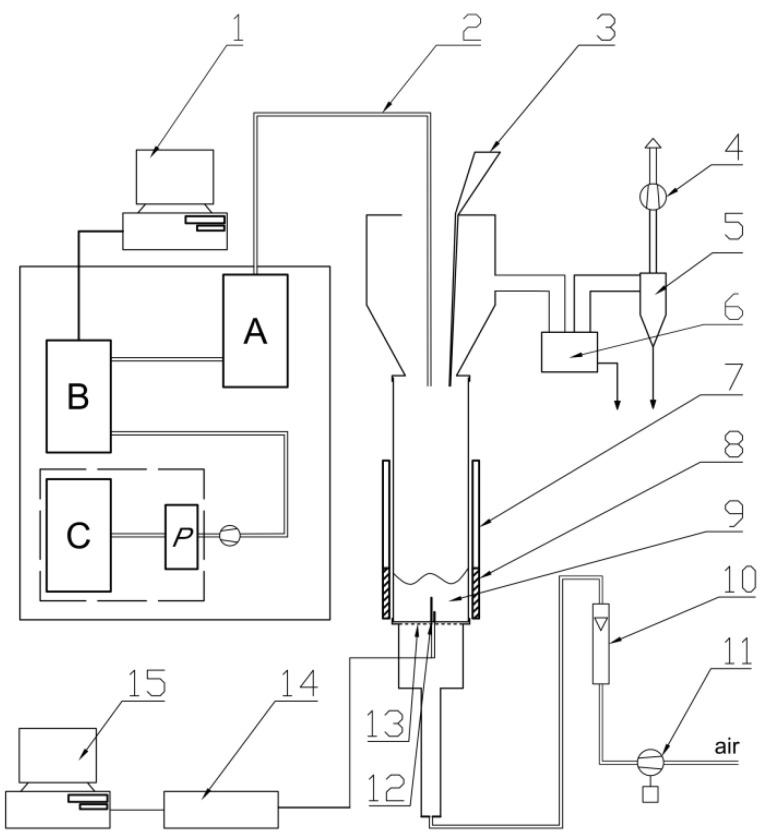
Scheme of the fluidized bed reactor: 1—computer; 2—heated probes for sampling the flue gases; 3—batcher; 4—exhaust fan; 5—cyclone; 6—ash trap for coarser particles; 7—movable radiation shield; 8—heating jacket; 9—bubbling bed; 10—air rotameter; 11—blower for fluidizing air; 12—two thermocouples; 13—flat, perforated metal plate distributor; 14—A/D convertor for thermocouple signals; 15—computer storing chemical analyses quantities and temperature; A—mobile conditioning system of Gasmet DX-4000; B—FTIR analyser (Gasmet DX-4000); C—Horiba PG250; P—Peltier’s cooler [23,24,25].

**Figure 2 polymers-15-01975-f002:**
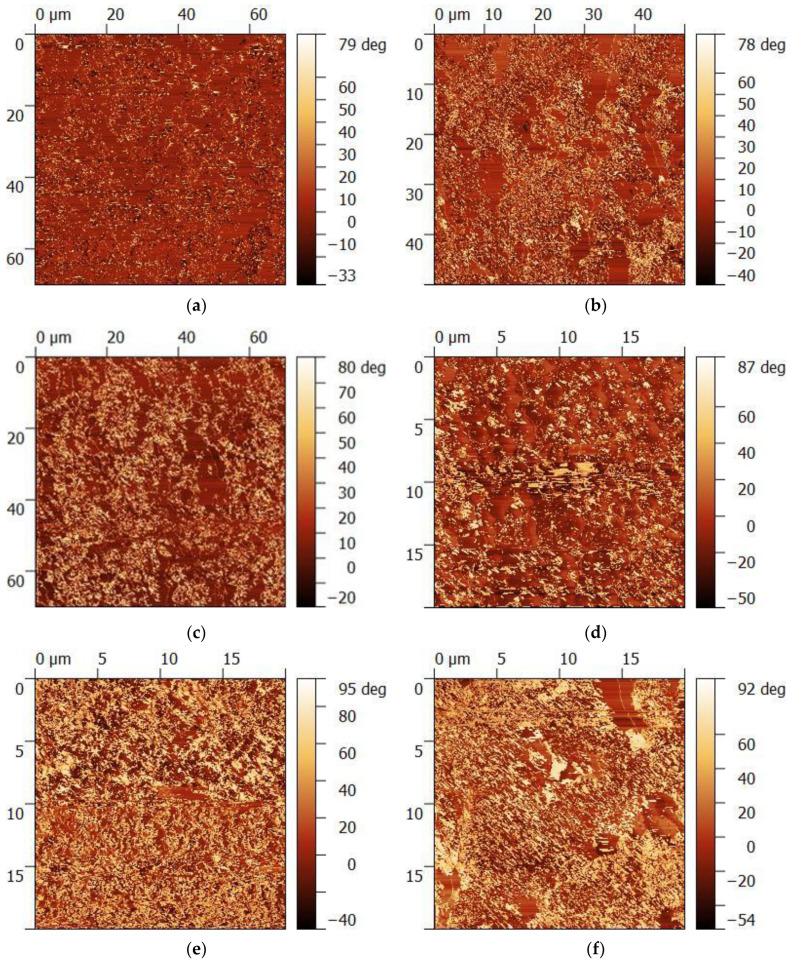
Distribution of natural fillers in the NR matrix: (**a**,**b**) CW; (**c**,**d**) CM; (**e**,**f**) CS.

**Figure 3 polymers-15-01975-f003:**
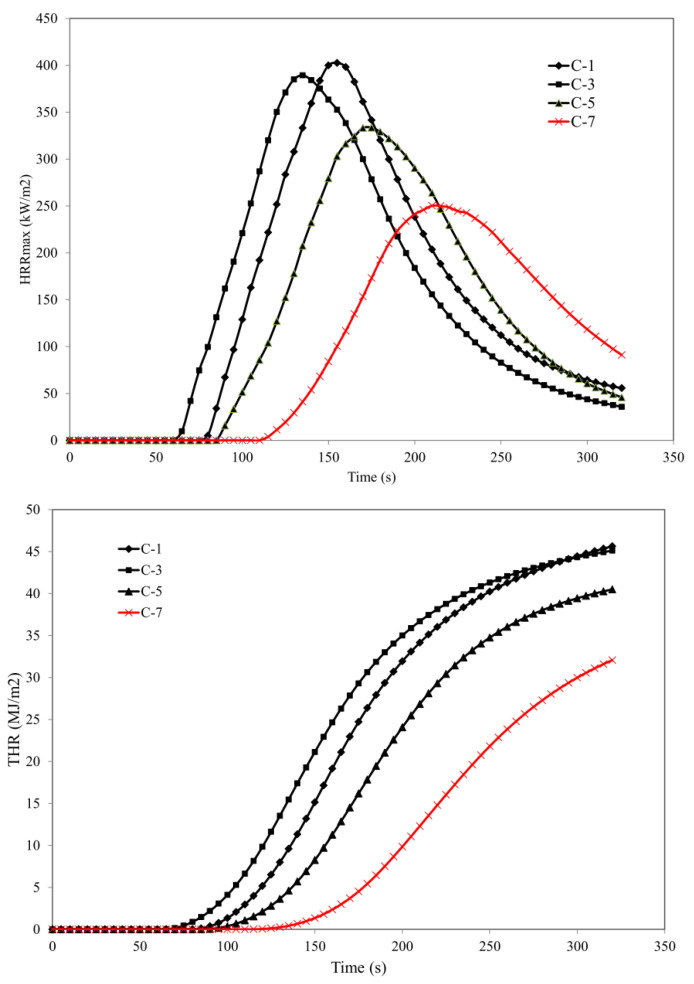
Flammability test results obtained using the cone calorimeter method.

**Figure 4 polymers-15-01975-f004:**
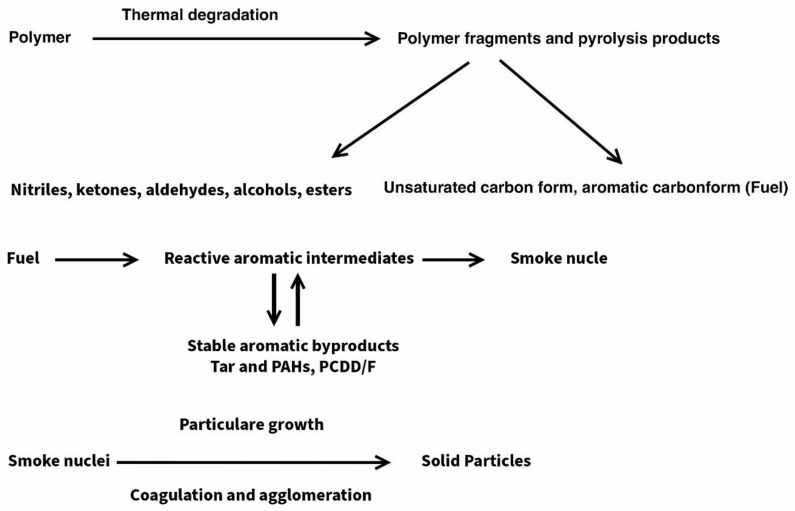
Mechanism of smoke formation [36,37].

**Figure 5 polymers-15-01975-f005:**
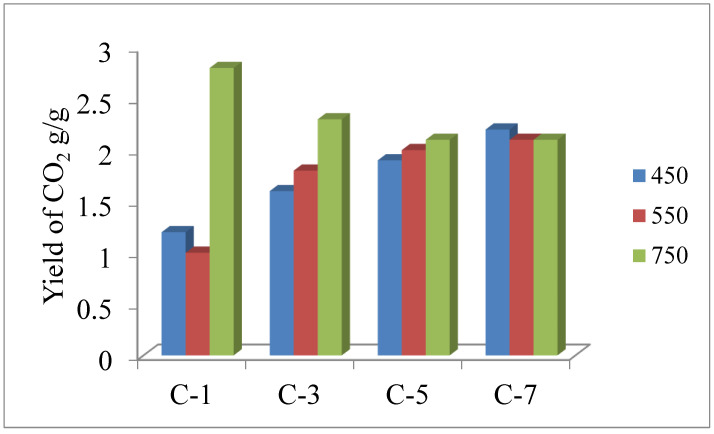
Values of specific emission of CO_2_, CO, SO_2_, and HCN.

**Table 1 polymers-15-01975-t001:** Composition of the NR composites (in phr—parts per hundred parts of rubber).

Composite	C-1	C-2	C-3	C-4	C-5	C-6	C-7
NR	100	100	100	100	100	100	100
S	2	2	2	2	2	2	2
MBT	2	2	2	2	2	2	2
SA	1	1	1	1	1	1	1
ZnO	5	5	5	5	5	5	5
CB	20	−	−	−	−	−	−
MPP	−	10	10	10	10	10	10
CS:CB (2:1)	−	20	−	−	−	−	−
CS:CB (5:1)	−	−	20	−	−	−	−
CM:CB (2:1)	−	−	−	20	−	−	−
CM:CB (5:1)	−	−	−	−	20	−	−
CW:CB (2:1)	−	−	−	−	−	20	−
CW:CB (5:1)	−	−	−	−	−	−	20

**Table 2 polymers-15-01975-t002:** Flammability results of NR rubber composites obtained using cone calorimetry method.

Sample	C-1	C-2	C-3	C-4	C-5	C-6	C-7
t_i_ (s)	76	53	58	47	66	75	91
t_f-o_ (s)	258	339	321	259	288	271	423
HRR (kW/m^2^)	229.3	146.9	170.8	197.4	183.4	187.4	111.8
HRR_max_ (kW/m^2^)	402.5	344.9	389.3	359.5	342.1	333.5	250.1
tHRR_max_ (s)	155	125	135	125	155	175	210
THR (MJ/m^2^)	30.8	31.8	35.1	31.1	30.1	27.1	27.0
EHC (MJ/kg)	26.7	20.3	21.3	20.6	19.1	17.8	16.2
EHC_max_ (MJ/kg)	41.4	49.3	49.1	44.2	48.1	47.2	43.9
MLR (g/s)	0.076	0.064	0.07	0.084	0.085	0.093	0.061
MLR_max_ (g/s)	0.254	0.229	0.293	0.237	0.228	0.238	0.212
AMLR (g/m^2^×s)	19.26	15.73	18.82	16.94	17.76	16.70	13.20
FIGRA (kW/m^2^s)	2.59	2.75	2.88	2.86	2.21	1.90	1.19
MARHE (kW/m^2^)	163.8	160.3	175.0	174.3	149.4	139.1	100.3

**Table 3 polymers-15-01975-t003:** Results of smoke emission during thermal decomposition of NR composites.

Sample	SD_MAX_	VOF4
C-1	451.7	707.4
C-2	471.1	667.4
C-3	436.6	551.7
C-4	434.4	592.0
C-5	402.0	536.8
C-6	398.5	609.4
C-7	399.2	505.0

**Table 4 polymers-15-01975-t004:** Concentration of PCDDs/Fs in gaseous thermal decomposition products of investigated composites of NR rubber.

Dioxin pg/g	C-1	C-3	C-5	C-7
2,3,7,8-Tetrachlorodibenzo-p-dioxin	0.5656	0.1016	0.1907	0.2491
1,2,3,7,8-Pentachlorodibenzo-p-dioxin	1.8094	<LOQ	<LOQ	1.5466
1,2,3,4,7,8-Hexachlorodibenzo-p-dioxin	0.2107	0.1092	0.0652	0.0205
1,2,3,6,7,8-Hexachlorodibenzo-p-dioxin	0.3737	0.1926	0.1150	0.0362
1,2,3,7,8,9-Hexachlorodibenzo-p-dioxin	0.1567	0.1291	0.0499	0.0448
1,2,3,4,6,7,8-Heptachlorodibenzo-p-dioxin	0.0231	<LOQ	<LOQ	<LOQ
Octachlorodibenzo-p-dioxin	0.001	0.0019	<LOQ	0.0013
2,3,7,8-Tetrachlorodibenzofuran	0.1786	0.0515	0.0923	0.1196
1,2,3,7,8-Pentachlorodibenzofuran	0.0185	<LOQ	<LOQ	0.0141
2,3,4,7,8-Pentachlorodibenzofuran	<LOQ	0.1076	0.0898	0.4281
1,2,3,4,7,8-Hexachlorodibenzofuran	0.0871	0.1450	0.0514	0.2387
1,2,3,6,7,8-Hexachlorodibenzofuran	0.0412	0.0686	0.0243	0.0767
2,3,4,6,7,8-Hexachlorodibenzofuran	<LOQ	0.0214	0.0247	<LOQ
1,2,3,7,8,9-Hexachlorodibenzofuran	0.1643	0.0891	<LOQ	0.1338
1,2,3,4,6,7,8-Heptachlorodibenzofuran	0.1898	0.0597	<LOQ	0.0806
1,2,3,4,7,8,9-Heptachlorodibenzofuran	0.1679	<LOQ	0.0838	0.1067
Octachlorodibenzofuran	0.0046	0.0033	<LOQ	<LOQ
Sum	3.9921	1.0806	0.7870	3.0969

**Table 5 polymers-15-01975-t005:** Values of TEF for PCDDs/Fs.

Congener PCDDs	TEF	Congener PCDFs	TEF
2,3,7,8-TCDD	1	2,3,7,8-TCDF	0.1
1,2,3,7,8-P_5_CDD	1	1,2,3,7,8-P_5_CDF	0.05
1,2,3,4,7,8-H_6_CDD	0.1	2,3,4,7,8-P_5_CDF	0.5
1,2,3,6,7,8-H_6_CDD	0.1	1,2,3,4,7,8-H_6_CDF	0.1
1,2,3,7,8,9-H_6_CDD	0.1	1,2,3,6,7,8-H_6_CDF	0.1
1,2,3,4,6,7,8-H_7_CDD	0.01	2,3,4,6,7,8-H_6_CDF	0.1
OCDD	0.0001	1,2,3,7,8,9-H_6_CDF	0.1
−	−	1,2,3,4,6,7,8-H_7_CDF	0.01
−	−	1,2,3,4,7,8,9-H_7_CDF	0.01
−	−	OCDF	0.0001

**Table 6 polymers-15-01975-t006:** TEQ values calculated for the investigated NR composites.

Dioxin pg/g	C-1	C-3	C-5	C-7
2,3,7,8-Tetrachlorodibenzo-p-dioxin	0.5656	0.1016	0.1907	0.2491
1,2,3,7,8-Pentachlorodibenzo-p-dioxin	1.8094	<LOQ	<LOQ	1.5466
1,2,3,4,7,8-Hexachlorodibenzo-p-dioxin	0.0211	0.0109	0.0065	0.0021
1,2,3,6,7,8-Hexachlorodibenzo-p-dioxin	0.0374	0.0193	0.0115	0.0036
1,2,3,7,8,9-Hexachlorodibenzo-p-dioxin	0.0157	0.0129	0.0050	0.0045
1,2,3,4,6,7,8-Heptachlorodibenzo-p-dioxin	0.0002	<LOQ	<LOQ	<LOQ
Octachlorodibenzo-p-dioxin	0.0000	0.0000	<LOQ	0.0000
2,3,7,8-Tetrachlorodibenzofuran	0.0179	0.0052	0.0092	0.0120
1,2,3,7,8-Pentachlorodibenzofuran	0.0009	<LOQ	<LOQ	0.0007
2,3,4,7,8-Pentachlorodibenzofuran	<LOQ	0.0538	0.0449	0.2141
1,2,3,4,7,8-Hexachlorodibenzofuran	0.0087	0.0145	0.0051	0.0239
1,2,3,6,7,8-Hexachlorodibenzofuran	0.0041	0.0069	0.0024	0.0077
2,3,4,6,7,8-Hexachlorodibenzofuran	<LOQ	0.0021	0.0025	<LOQ
1,2,3,7,8,9-Hexachlorodibenzofuran	0.0164	0.0089	<LOQ	0.0134
1,2,3,4,6,7,8-Heptachlorodibenzofuran	0.0019	0.0006	<LOQ	0.0008
1,2,3,4,7,8,9-Heptachlorodibenzofuran	0.0017	<LOQ	0.0008	0.0011
Octachlorodibenzofuran	0.0000	0.0000	<LOQ	<LOQ
TEQ	2.5010	0.2366	0.2787	2.0794

**Table 7 polymers-15-01975-t007:** Concentration of PAHs in the gaseous thermal decomposition products of NR composites.

PAH µg/g	C-1	C-3	C-5	C-7
Naphthalene	18.35	5.47	0.63	0.68
2-Methylnaphthalene	61.45	14.21	4.11	7.48
1-Methylnaphthalene	48.95	15.03	4.27	7.21
Acenaphthylene	<LOQ	3.98	2.93	3.90
Acenaphthene	25.27	6.66	9.62	6.10
Fluorene	113.84	28.47	22.06	24.20
Phenanthrene	55.88	10.90	12.96	11.25
Anthracene	61.33	0.60	1.82	1.36
Fluoranthene	33.90	3.32	3.71	3.77
Pyrene	125.30	7.34	10.23	8.82
Benz(a)anthracene	14.60	1.06	0.80	1.15
Chrysene	8.97	1.74	1.13	1.93
Benzo(b)fluoranthene	5.61	1.07	0.61	0.71
Benzo(k)fluoranthene	4.71	0.81	0.50	0.63
Benzo(a)pyrene	6.63	1.08	0.69	0.83
Indeno(1,2,3-cd)pyrene	11.07	1.17	0.89	0.75
Dibenz(a,h)anthracene	9.08	1.00	0.92	0.57
Benzo(g,h,i)perylene	38.52	2.92	3.33	2.16
Sum	643.56	106.93	81.28	83.58

**Table 8 polymers-15-01975-t008:** Limit concentrations of gaseous components according to PN-88/B-02855 (g/g).

Gas	CO_2_	CO	NO_2_	SO_2_	HCl	HCN
LC${}_{50}^{30}$	194.4	3.75	0.205	0.7	1	0.16

**Table 9 polymers-15-01975-t009:** Toxicometric indices for composites of NR rubber.

Sample	T, °C	W_LC50_, g/m^3^						W_LC50M_, g/m^3^	W_LC50SM_, g/m^3^
		CO_2_	CO	NO_2_	SO_2_	HCl	HCN		
C-1	450	162 ± 14	27 ± 2	<LOQ	483 ± 66	<LOQ	<LOQ	22.2 ± 0.9	32.77
	550	200 ± 19	13 ± 1	<LOQ	315 ± 7	<LOQ	305 ± 27	11.7 ± 1.4	
	750	70 ± 1	3288 ± 514	<LOQ	1323 ± 315	<LOQ	2165 ± 477	64.4 ± 2.8	
C-3	450	125 ± 8	44 ± 1	<LOQ	407 ± 59	<LOQ	300 ± 21	28.1 ± 0.8	38.60
	550	110 ± 8	12 ± 1	<LOQ	253 ± 34	<LOQ	62 ± 3	8.6 ± 0.6	
	750	86 ± 3	4936 ± 1138	<LOQ	4021 ± 20	<LOQ	1663 ± 310	79.0 ± 2.1	
C-5	450	105 ± 5	43 ± 2	<LOQ	981 ± 187	<LOQ	184 ± 15	25.6 ± 0.9	39.60
	550	99 ± 7	12 ± 1	<LOQ	223 ± 31	<LOQ	53 ± 1	8.6 ± 0.3	
	750	93 ± 5	4596 ± 1108	<LOQ	1560 ± 608	<LOQ	2098 ± 571	84.6 ± 6.9	
C-7	450	91 ± 10	39 ± 8	<LOQ	584 ± 105	<LOQ	122 ± 10	21.7 ± 2.7	31.75
	550	93 ± 7	13 ± 1	<LOQ	125 ± 17	<LOQ	50 ± 3	8.6 ± 0.7	
	750	95 ± 1	3134 ± 758	<LOQ	284 ± 47	<LOQ	1117 ± 123	65.0 ± 2.0	

**Table 10 polymers-15-01975-t010:** Mechanical properties of NR composites.

Composite	SE_100_ MPa	σSE_100_ MPa	SE_200_ MPa	σSE_200_ MPa	SE_300_ MPa	σSE_300_ MPa	TS MPa	σTS MPa	Eb %	σEb %
C-1	0.89	0.06	1.62	0.09	2.94	0.17	15.00	0.47	603.2	16.1
C-2	1.08	0.02	1.88	0.05	2.78	0.07	13.85	0.03	607.5	27.8
C-3	1.26	0.05	2.23	0.08	3.27	0.16	15.20	0.98	527.9	98.6
C-4	0.78	0.03	1.19	0.04	1.90	0.13	16.90	0.40	640.3	5.7
C-5	1.22	0.08	2.10	0.22	2.97	0.33	15.40	0.87	682.4	57.2
C-6	0.96	0.17	1.75	0.45	2.66	0.74	15.84	0.76	477.0	55.0
C-7	1.01	0.05	1.88	0.10	2.82	5.44	13.20	1.10	652.0	13.7

## Data Availability

Data sharing not applicable.

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
