# Peer review of "Effect of Hybrid Filler, Carbon Black–Lignocellulose, on Fire Hazard Reduction, including PAHs and PCDDs/Fs of Natural Rubber Composites"

_polymers, 2023, doi:10.3390/polym15081975_

Round 1
Reviewer 1 Report
It is a valuable work to investigate the smoke emission, the toxicity of gaseous decomposition products of elastomeric composites. The manuscript can be accepted after minor revision.
1. If there is MPP composition in composite C-1, it will be clearer to elucidate the role of lignocellulose filler by comparing composites C2-7 with C-1.
2. Figure 3. Mechanism creating of smoke can be deleted.
3. Summary part: The authors should present the obtained important data results of tests of smoke emission, Dioxins Furans (PCDDs/Fs) and PAH toxicity, etc.
Reviewer 2 Report
Comments to Authors:
Ref. No.: polymers-2307182
Title: Effect of Hybrid Filler, Carbon Black-Lignocellulose on Fire Hazard Reduction, Including PAH and PCDDs/Fs of Natural Rubber Composites
Overview and general recommendation:
In this manuscript, the authors studied how the addition of lignocellulose fillers affects the fire hazards and smoke emission properties of carbon black-containing elastomeric composites. In addition, their mechanical properties were also evaluated. However, although it is very essential, in this study, it lacks a study about the properties and morphologies of those lignocellulose fillers added to the composites. The presentation in the manuscript is a mess. Some paragraphs are not complete. Some paragraphs are too short. They are not well-balanced. The writing is too abundant, and the authors really need to condense their writing, which can help improve the manuscript's readability.
Major comments:
1. First thing first, the properties and morphologies of fillers added to the composite will significantly affect its final properties, including fire hazards and mechanical properties. For example, nanocomposites will have different properties from traditional composites. So, the authors need to provide more information about the properties of their lignocellulose fillers, for example, purity, their sizes, and how they look like, etc.
2. In Fig. 2, the presentation of the fire hazards is not complete. What criteria did the authors use to define the flame out in the cone calorimeter tests? If we look at Fig. 2, at around 320 s, there is still some heat release, which means there is still some flame. For all samples, their heat release rates are different. So how do the authors define the flame out?
3. In all those figures in the manuscript, why only were C-1, C-3, C-5, and C-7 presented?
4. It is unclear why the authors think chlorine-based dibenzodioxins are a big concern in this study. The authors didn’t explain which component has a high loading of chlorine and where chlorine is from.
5. At the beginning of the section on smoke emission, the first 6 paragraphs are some introductory information to smoke, which is good to know but not necessary for a scientific study. There are many other examples like this. The writing is too abundant, and the authors really need to condense their writing.
6. When the authors explained how lignocellulose fillers could reduce the fire hazards of elastomeric composites, some figures of the char and their SEM images will significantly help this.
7. There are still many grammar errors in the manuscript.
Reviewer 3 Report
Paper abstract. The smoke emitted during thermal decomposition of elastomeric composites contains significant number of carcinogenic and mutagenic compounds from the group of polycyclic aromatic hydrocarbons, PAHs, as well as polychlorinated dibenzo-p-dioxins and furans, PCDDS/Fs. By replacing carbon black with a specific amount of lignocellulose filler, we may noticeably reduce the fire hazard caused by elastomeric composites. The lignocellulose filler reduces the parameters associated with the flammability of the tested composites, decreases the smoke emission, as well as limits the toxicity of gaseous decomposition products expressed as a toximetric indicator and the sum of PAHs and PCDDs/Fs. The natural filler also reduces emission of gases that constitute the basis for determination of the value of toximetric indicator WLC50SM. The flammability and optical density of the smoke were determined in accordance with the applicable European standards, with the use of cone calorimeter and a chamber for smoke optical density tests. PCDD/F and PAH were determined using GC-MS-MS technique. The toximetric indicator was determined using FB-FTIR method (fluidised bed reactor and the infrared spectrum analysis).
1. Some word are Polish in the paper. E.g. "Próbka" in the Table 9.
2. Except of Figure. 4, the time dependence of CO2, CO, SO2 and HCN should be added to the manuscript.
3. Specific Emission of CO2, CO, etc... should be replaced by term yield of CO2, CO, etc.
4. Table 8 (the unit of concentration is missing).
5. The correlation between CO and CO2 would be very interesting.
